# Converging photospheric vortex flows close to the polarity inversion line of a fully emerged active region

Jean C. Santos[1] and Cristiano M. Wrasse[2]

[1]Universidade Tecnológica Federal do Paraná, Curitiba-PR, Brazil.
[2]Instituto Nacional de Pesquisas Espaciais, São José dos Campos-SP, Brazil.

**Correspondence:** Jean C. Santos (jeansantos@utfpr.edu.br)

**Abstract.** We report on the occurrence of vortexes in flow fields obtained from the evolution of the line-of-sight component of the photospheric magnetic field in a region around the polarity inversion line (PIL) of a fully emerged active region. Based on a local linear approximation for the flow field, we identify the presence of critical points and classify them according to the eigenvalues of the Jacobian matrix of the linear transformation. Converging vortexes are associated to the presence of a particular kind of critical point, known as Attracting Focus. We identified twelve converging vortexes in the analyzed period and detected the occurrence of other types of critical points, which indicate the complexity of the flow field around the PIL. The detected vortexes show a clockwise preferred sense of rotation with approximately 67% of the cases. A geometrical analysis of the velocity structures produced an average value of $\bar{D} = 1.63 \pm 0.05$ for the fractal dimension, which is very close to the one obtained for isotropic homogeneous turbulence ($D = 5/3$). This suggests that the flow around the PIL is turbulent in nature.

## 1 Introduction

Horizontal flow fields in the solar photosphere have an important role in the concentration and dispersal of surface magnetic flux. Just to give some examples, surface flows are responsible for magnetic flux concentration at the border of the convection cells (magnetic network); they disperse the magnetic flux of active regions (turbulent diffusion); and they transport the magnetic flux to the poles (meridional flow). Horizontal flows may also contribute to magnetic energy and helicity injection into the upper atmosphere by twisting and interweaving the footpoints of flux-tubes, generating field aligned currents and magneto-hydrodynamic waves, and also may be responsible for the occurrence of magnetic reconnection by bringing together opposite magnetic polarity regions. Therefore, the investigation of photospheric horizontal motion patterns responsible for the evolution of magnetic features in the solar photosphere may give some clues to understand how the combination of these two quantities, magnetic field and flow field, influence the solar activity.

In this sense, vortex/rotational motion patterns are particularly important for solar activity. In the quiet Sun convective flows concentrate magnetic fields in the downdraft regions of the convective cells. The conservation of angular momentum forces

the plasma to rotate around the center of the downdraft, generating small scale vortexes. These vortexes have been extensively detected in observations of the photosphere (Brandt et al., 1988; Simon and Weiss, 1997; Attie et al., 2009; Bonet et al., 2010; Balmaceda et al., 2010; Vargas Domínguez et al., 2011) and also have been observed in the quiet Sun chromosphere as a signature of plasma moving along curly magnetic field lines in coronal holes (Wedemeyer-Böhm and Rouppe van der Voort, 2009). Simulations indicate that the vortexes occurring in strongly magnetized regions are closely connected with dissipation processes providing localized heating in the lower parts of the solar atmosphere (Moll et al., 2012) and observations usually associate vortex detection with bright points (Bonet et al., 2008).

On larger scales, rotational motions were observed in sunspots and they are usually associated with energy and helicity build-up and later release by flare and/or coronal mass ejection (Brown et al., 2003; Hiremath and Suryanarayana, 2003; Hiremath et al., 2005; Yan and Qu, 2007; Yan et al., 2008; Min and Chae, 2009; Yan et al., 2009; Kazachenko et al., 2010; Zhu et al., 2012; Yan et al., 2012; Jiang et al., 2012; Vemareddy et al., 2012; Hardersen et al., 2013; Wang et al., 2014; Ruan et al., 2014; Gopasyuk, 2015; Li and Liu, 2015; Suryanarayana et al., 2015; Wang et al., 2016; Vemareddy et al., 2016). The rotation of sunspots is usually very slow, this means that the evolution of the magnetic field in the corona associated to it would be slow as well. However, strong flares (M- or X-class) are sometimes associated to rapid (abnormal) sunspot rotation. At the moment there is not a defined mechanism to explain sunspot rotation. It is suggested that it could be a result of the interaction of the flux tube with photospheric flows, during or after its emergence, or the effect of the emergence of a twisted flux tube. Changes of rotational pattern of sunspot after the flare occurrence were also observed (Liu et al., 2016; Bi et al., 2016) and they are associated to Lorentz forces.

In this work, we investigate the properties of the flow field obtained from the evolution of photospheric magnetic features around the polarity inversion line (PIL) of a fully emerged Active Region (AR). We targeted the PIL since it is the place where opposite polarity magnetic fields interact and where sharp changes are usually associated to the onset of flares (Severnyi, 1958; Wang et al., 1994; Kosovichev and Zharkova, 2001; Sudol and Harvey, 2005; Sharykin et al., 2017). We first focus in the detection and classification of critical points. Critical points are points where the velocity vanishes and their importance resides in the fact that the flow may be directed to and rotate around these points forcing opposite polarities to meet and annihilate there, contributing to the energetics of the solar atmosphere. The critical points are used to identify the presence of converging vortex flows in the region around the PIL. Finally, we investigate the geometric structure of the flow, by calculating its the fractal dimension, and use it as an indicative that the flow around the PIL presents turbulent nature.

## 2  Data and methodology

We selected as our target the AR NOAA 9289, located at the southern solar hemisphere. Figure 1 shows the full disk line-of-sight (LOS) component of the photospheric magnetic field (top panel) and a close view of AR NOAA 9289 (bottom panel), as measured by the Michelson Doppler Imager (MDI) (Scherrer et al., 1995) on January $2^{nd}$ 2001 at 04:51:01 UT. The MDI instrument obtain images of the full disk of the Sun using a 1024x1024 pixel CCD camera, with a spatial resolution of 2 arcsec

per pixel and a temporal resolution of 96 minutes. The noise level of the instrument is about 7.6 gauss in the 96 min full disk mode. In Fig. 1, the LOS magnetic field ($B_{LOS}$) is saturated at $\pm 100$ G for a better visualization of the magnetic features.

The AR consisted of a large bipolar magnetic field with a leading negative polarity and a following positive polarity region. From December $31^{st}$ 2000 to January $3^{rd}$ 2001 the region was fully emerged and its leading sunspot was seen to rotate about 50° clockwise with an average speed of 0.56° $h^{-1}$ (Zhu et al., 2012). This clockwise rotation may cause a shear in the polarity inversion line, increasing the energy and relative helicity flow out of the photospheric plane around that region. As shown in Figure 1, there was a smaller bipolar AR close to AR NOAA 9289.

We focus in a region of approximately $54 \times 78$ arcsec$^2$ around the PIL and follow the evolution of $B_{LOS}$ for a period of 3.6 days starting on December $31^{st}$ 2000 at 01:34:43 UT. Figure 2 display the $B_{LOS}$ measurements with a cadence of 192 minutes. The x and y axis show the spatial coordinates in pixel values, where each pixel corresponds approximately to 1.2 arcsec. During this period, AR NOAA 9289 crossed the center of the solar disk and the LOS component of the photospheric magnetic field is considered identical to component perpendicular to the photosphere. In case the AR is far from the disk center, a correction would be necessary to find the perpendicular component, since the LOS component does not correspond to the perpendicular component anymore. Visual inspection shows that initially the magnetic field around the PIL is very fragmented with very small positive and negative polarity regions randomly distributed. These fragmented polarities start to coalesce forming a negative polarity region (N1), connected to the AR main negative polarity, and two smaller positive polarity regions, one northern of the negative polarity (P1) and other southern (P2). The positive polarities P1 and P2 coalesce with two smaller positive polarity regions that were located around them, indicated by red arrows in the figure. Later, the positive polarity P2 connects to the AR main positive polarity. During this process, it seems that the negative polarity (N1) is deforming while the positive polarity (P1) is protruding into it, which could be a triggering mechanism for flare occurrence (Kusano et al., 2012; Toriumi et al., 2013). Finally, the negative polarity N1 starts to rotate clockwise around itself and the positive polarity P1 starts to rotate in the same sense around N1, moving between N1 and the AR leading negative polarity. As discussed in Toriumi et al. (2013) for AR NOAA 11156, the evolution of the small scale magnetic field around the PIL may increase the shear and contribute to the triggering of strong flares.

To determine the velocity field responsible for the changes observed in the $B_{LOS}$ component of the magnetic field, we have used the Local Correlation Tracking (LCT) technique (November and Simon, 1988). More specifically, we have used a Fourier-based local correlation tracking (FLCT) implementation described in Welsch et al. (2004). The LCT/FLCT velocity indicates the apparent movement of the footpoint of the magnetic field lines. If we assume that the photospheric field is vertical due to the buoyance of magnetic flux tubes, then a sequence of images will show the horizontal motion of the footpoints of the magnetic flux tubes. The velocity is locally determined by cross correlating a small fraction of two subsequent images shifted by a variable displacement. The shift having the highest correlation shows the relative displacement and the tracking velocity is obtained by dividing this displacement by the time interval between two images of a sequence. This velocity is used as a proxy for the tangential plasma velocity.

Unfortunatelly, there are some issues regarding the LCT/FLCT method. As pointed by Démoulin and Berger (2003), if the flux tube is inclined when it rises through the solar atmosphere, the point where it crosses the photosphere moves horizontally.

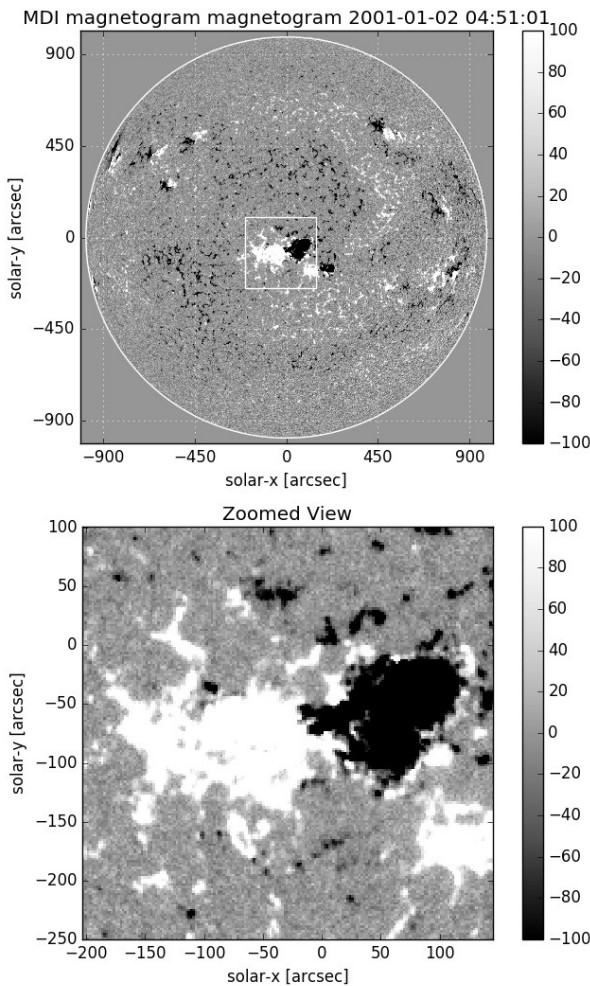

**Figure 1.** Full disk LOS magnetic field (top panel) and close view of AR NOAA 9289 (bottom panel) as measured by the MDI/SoHO instrument on Januray $2^{nd}$ 2001 at 04:51:01 UT.

This apparent motion can be interpreted by the LCT/FLCT method as a proper horizontal motion and Démoulin and Berger (2003) suggested a correction, if the tangential component of the magnetic field and the vertical component of the velocity are known. Since in our case the AR is not emerging, i.e. no vertical motions, this correction is not necessary. Another issue, pointed by Schuck (2005), is that LCT/FLCT does not permit any contraction, dilation, or rotation of the magnetic fluid on the scale of the apodizing window. Schuck (2005, 2008) suggests to combine the LCT/FLCT with the difference affine method, in a method called Difference Affine Velocity Estimator (DAVE), to account for the convergence and divergence in the flow, as well

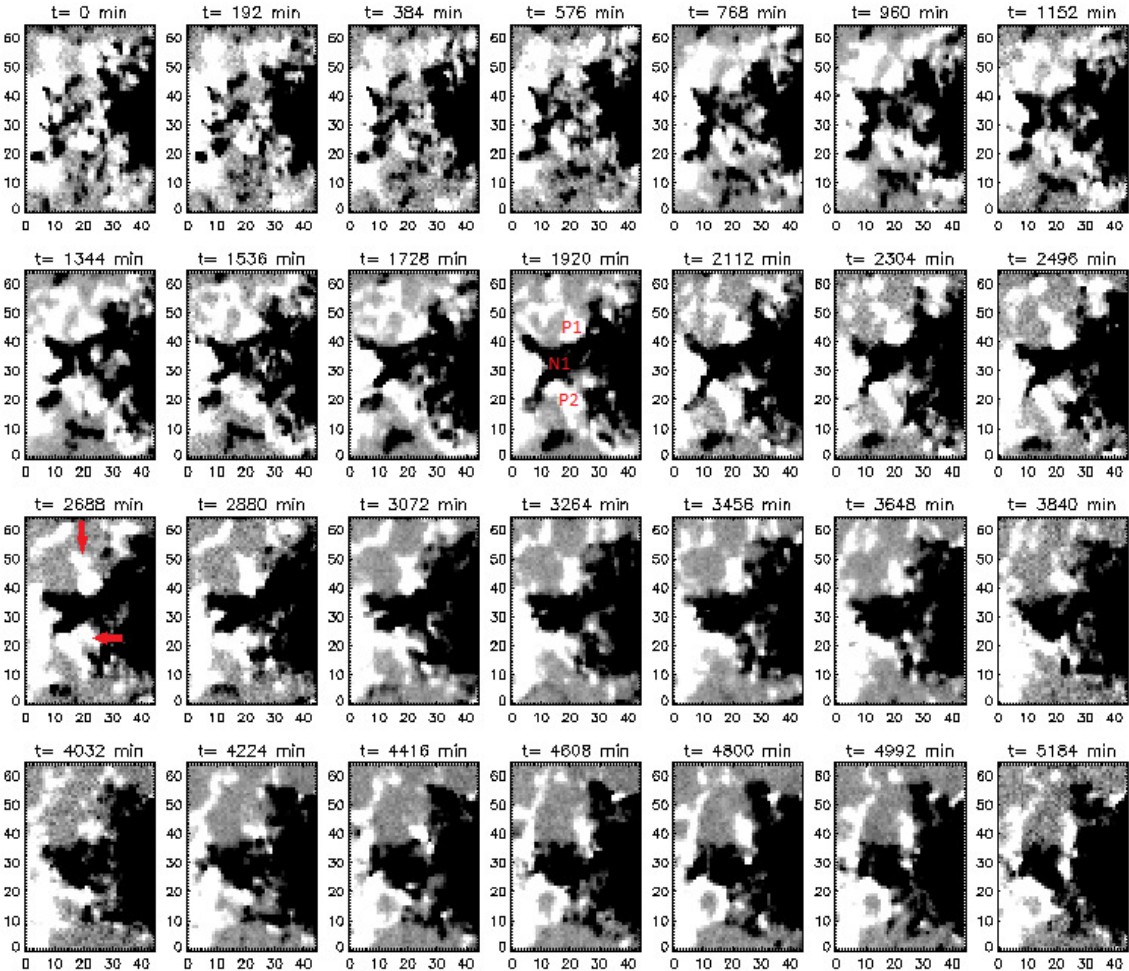

**Figure 2.** Evolution of the LOS component of the magnetic field around the PIL for the period of 3.6 days starting on December $31^{st}$ 2000 at 01:34:43 UT.The x and y axis show the spatial coordinates in pixel values, where each pixel corresponds approximately to 1.2 arcsec.

as higher order parametric profiles. Finally, the LCT/FLCT method considers that the vertical component of the magnetic field evolves according the advection equation, while in fact it evolves according the induction equation. Different methods to obtain the horizontal velocity from the evolution of the normal component of the magnetic field that obey the induction equation are available (Welsch et al., 2004; Longcope, 2004; Schuck, 2008). However, they usually require extra information about the horizontal component of the magnetic field and the vertical component of the velocity, or they impose some restrictions to the velocity field.

For the sake of simplicity, and because we have information only about the temporal evolution of $B_{LOS}$, we decided to use LCT/FLCT method even knowing about the issues that the application of the method imply. We use the time of the sample of the $B_{LOS}$ component ($\approx$192 min) and select a full width half maximum (FWHM) window of 5 pixel ($\approx$6 arcsec) to perform

the localized cross-correlation. In applying the LCT/FLCT method we have considered $B_{LOS}$ as a passive scalar and have assumed that all the changes observed in Figure 2 are due the horizontal displacement of the magnetic features, with no flux emergence or submergence.

## 2.1 Critical point detection and classification

Photospheric vortexes are important since they can form twisted flux tubes, transporting magnetic energy and helicity to upper layers of the solar atmosphere in the process. They are usually detected by visual inspections and, since this may cause a bias in the analysis of vortexes, efforts in developing automated detection methods have been recently performed (Kato and Wedemeyer, 2017; Rempel et al., 2017; Giagkiozis et al., 2018). In this work, we intend to contribute to the development of automated methods of vortex detection by using a well known method of detection and classification of critical points (Helman and Hesselink, 1989) to identify converging vortexes by associating them to a specific kind of critical point.

In a 2D flow field the velocity vector is given at any point as $\boldsymbol{v}(x,y) = v_x(x,y)\hat{i} + v_y(x,y)\hat{j}$. If we consider a linear vector field approximation, the velocity vector components can be written in terms of the $(x,y)$ coordinate components as

$$
\begin{pmatrix} v_x(x,y) \\ v_y(x,y) \end{pmatrix} = \begin{pmatrix} a & b \\ d & e \end{pmatrix} \begin{pmatrix} x \\ y \end{pmatrix} + \begin{pmatrix} c \\ f \end{pmatrix} \tag{1}
$$

where

$$
\mathbf{J} = \begin{pmatrix} a & b \\ d & e \end{pmatrix} \tag{2}
$$

is the Jacobian matrix of the transformation.

So, to represent the velocity vector in a linear approximation we need to know the values of the constants $a$, $b$, $c$, $d$, $e$ and $f$. To find the values of the constants it is necessary to consider the velocity vector in at least three points around the region of interest (ROI), in order to solve the following linear system of equations

$$
\begin{bmatrix} x_1 & y_1 & 1 \\ x_2 & y_2 & 1 \\ x_3 & y_3 & 1 \end{bmatrix} \times \begin{bmatrix} a \\ b \\ c \end{bmatrix} = \begin{bmatrix} v_x(x_1,y_1) \\ v_x(x_2,y_2) \\ v_x(x_3,y_3) \end{bmatrix} \tag{3}
$$

for $a$, $b$, $c$, and the linear system of equations

$$
\begin{bmatrix} x_1 & y_1 & 1 \\ x_2 & y_2 & 1 \\ x_3 & y_3 & 1 \end{bmatrix} \times \begin{bmatrix} d \\ e \\ f \end{bmatrix} = \begin{bmatrix} v_y(x_1,y_1) \\ v_y(x_2,y_2) \\ v_y(x_3,y_3) \end{bmatrix} \tag{4}
$$

for $d$, $e$ and $f$. We solve the set of simultaneous linear equations of the form $Ax = b$ by back-substitution using the IDL functions SVDC and SVSOL.

**Table 1.** Classification of critical points according to the values of the real (R) and imaginary (I) parts of the eigenvalues.

| Critical point type | Real part of the eigenvalues | Imaginary part of the eigenvalues |
|---|---|---|
| Saddle point | $R1 < 0, R2 > 0$ | $I1 = I2 = 0$ |
| Attracting node | $R1, R2 < 0$ | $I1 = I2 = 0$ |
| Repelling node | $R1, R2 > 0$ | $I1 = I2 = 0$ |
| Attracting focus | $R1 = R2 < 0$ | $I1 = -I2 <> 0$ |
| Repelling focus | $R1 = R2 > 0$ | $I1 = -I2 <> 0$ |
| Center | $R1 = R2 = 0$ | $I1 = -I2 <> 0$ |

Once the linear representation of the field is available, we can use Equation 1 to check if there is a critical point inside the ROI. Critical points may be interpreted as the fixed points of a map. Given the location of such a points and their types, the behaviour of the orbit of the particles can be predicted around them. Also, critical points are the only points where the flow field lines are allowed to instersect. A critical point is characterized by a flow velocity given by $\boldsymbol{v}(x,y) = (0,0)_{ij}$. Then, in a linear vector field approximation, to find the coordinates of the critical point we have to solve a matrix equation like

$$\begin{bmatrix} a & b \\ d & e \end{bmatrix} \times \begin{bmatrix} x \\ y \end{bmatrix} + \begin{bmatrix} c \\ f \end{bmatrix} = \begin{bmatrix} 0 \\ 0 \end{bmatrix} \tag{5}$$

for $x$ and $y$. The solution of this equation gives the $(x,y)$ coordinates of the critical point, in case it exists.

From the eigenvalues of the Jacobian matrix, given by the solution of

$$\left\| \begin{bmatrix} a & b \\ d & e \end{bmatrix} - \lambda \begin{bmatrix} 1 & 0 \\ 0 & 1 \end{bmatrix} \right\| = 0, \tag{6}$$

we can classify the critical point, according to Helman and Hesselink (1989), as presented in Table 1. We find the eigenvalues by first reducing the Jacobian matrix to upper Hessenberg form using the ELMHES function in IDL and then returning the eigenvalues by applying the HQR function.

To automatically search for critical points, we scan the 2D LCT/FLCT vector field using a rectangle of size $(\Delta x, \Delta y)$. The left panel on Figure 3 shows an illustration of the process for the vector field obtained at t=768 min. Using the information of the flow and the coordinates at three corners of the rectangle, we perform the calculations described previously to search for critical points and classify them. By choosing different corners we cover the whole area inside the rectangle and after the calculation is finished we move the rectangle in the x direction by $\Delta x$ and start the calculations again, until the end of the line is reached. We then go to the next line by moving the rectangle $\Delta y$ in the vertical direction until the complete 2D flow field is covered. The right panel on Figure 3 shows the result of this calculation where the circles indicate the position of the suggested critical points. The different colors indicate the classification of the critical points: blue–Saddle Point, red–Attracting Node/Focus and yellow–Repelling Node/Focus. The solid (dashed) contour line indicate the regions where $B_{LOS}$ equals the value of +100 G (-100 G). The results are sensitive to the size of the rectangle and should be cross checked by visual inspection

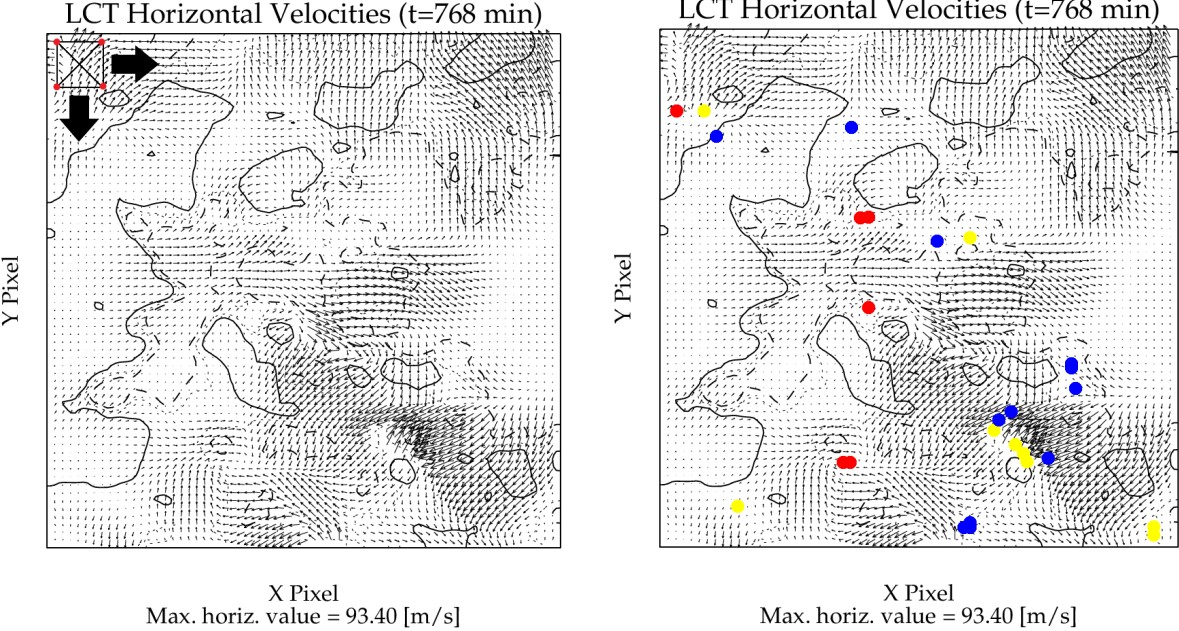

**Figure 3.** Left panel - Illustration of the method for automatic search of critical points in a 2D flow field. Right panel - results obtained using the automatic search for the LCT flow field at t=768 min. The circles show the location of the suggested critical points and the colors show their classification (blue: Saddle Point, red: Attracting Node/Focus, yellow: Repelling Node/Focus).

since the calculations may produce false positive cases or even miss some critical points. This cross check may be performed with the original 2D flow field, a re-normalized one or any visualization that could facilitate the identification of the critical points like the line integral convolution (LIC) technique, for example.

## 2.2 Fractal dimension and the box couting method

5 A fractal is defined as a set for which the Hausdorff Besicovitch dimension (fractal dimension) exceeds the topological dimension (Mandelbrot, 1982). To calculate the fractal dimension of an image we use the box couting method. In this method, an image is covered by a sequence of grids of decreasing sizes and for each of the grids we compute the number of square boxes intersected by the image, $N(s)$, and the side length of the squares, $s$. The regression slope $D$ of the straight line formed by plotting $\log(N(s))$ against $\log(1/s)$ indicates the degree of complexity, or fractal dimension, between 1 and 2 ($1 \leq D \leq 2$).

10 $$D = \frac{\log N(s)}{\log(1/s)} \tag{7}$$

Mandelbrot (1975) suggested that turbulent shapes require a proper geometrical description. For isoscalar surfaces in a 3D homogeneous turbulence, he argued for a fractal dimension $D = 8/3$, which corresponds to $D = 5/3$ in 2D, if turbulence could be described as possessing "Kolmogorov-Gauss" scaling (Mandelbrot, 1975, 1982).

## 3 Results

We apply the method described in Section 2 to identify and classify the critical points in the data cube containing the 2D LCT/FLCT flow fields for the 3.6 days period, starting on December $31^{st}$ 2000 at 01:34:43 UT. Before we apply the method, each flow field is re-sampled to have $128 \times 128$ data points. We use a rectangle of $2 \times 2$ to scan the flow field since this is the resolution necessary to detect the smallest structures in the flow field. We select only the critical points classified as Attracting Focus and crosscheck the results with a flow field normalized in a way that all the flow vectors have the same size. These critical points are associated to vortex flows converging to them.

Figure 4 shows the 2D flow fields and the positions (red circles) where the presence of converging vortex flows were confirmed. We identified the occurrence of 12 (twelve) converging vortexes in the LCT/FLCT flow field obtained from the evolution of the LOS photospheric magnetic field around the PIL, for a period of 3.6 days. The arrows in Figure 4 show the direction of rotation, with about 67% (8) of the cases rotating clockwise and 33% (4) counterclockwise. This shows a preference to clockwise rotation in the set of converging vortexes detected around the PIL.

Our investigation also shows that critical points are always present in the LCT/FLCT flow fields for the period analyzed. Their total number varies with time and Saddle Points are the most commonly detected type of critical point, with a total of 213 detected in the period of 3.6 days. Figure 5 shows a rough estimation of the noncumulative number of critical points computed for each time instant analysed. This result probably reflects the complexity of the flow field around the PIL since the lines connecting the critical points (separatrices) separate different flow regions. The detection method produced an accuracy of approximately 70% in finding the critical points in the complex velocity fields obtained by LCT/FLCT.

We also investigate the geometric aspect of the flow by calculating its fractal dimension. It describes how detail in a pattern changes with the scale at which is measured and provides a measure of geometrical complexity (Mandelbrot, 1982). To perform this calculation we apply a mask to the flow field, selecting only the regions where the velocity amplitude is larger than $v = 23.6$m/s. Figure 6 shows the time evolution of the distribution of the regions where the velocity amplitude is above the threshold, shown in black. We want to measure the fractal dimension of those structures. The selected threshold corresponds to the average of the velocities presented in the top panel of Figure 7.

The fractal dimension is calculated using the box counting method described in Section 2. The bottom panel of Figure 7 shows the results obtained for each time instant. They result in an average fractal dimension of $\bar{D} = 1.63 \pm 0.05$. This fractal dimension is very close to the one obtained for homogeneous turbulence ($D = 5/3$), suggesting the occurrence of a turbulent flow around the PIL. Since fully developed turbulence consists of a hierarchy of eddies, we expect that vortex flows will be a common feature of the flow field around the PIL.

## 4 Conclusions

We have investigated the LCT/FLCT flow fields obtained from the evolution of $B_{LOS}$ in a region around the PIL for the presence of converging vortex flows. To perform this, we first look for the presence of critical points, using a linear approximation of the flow field, and classify them according to the eigenvalues of the Jacobian matrix of the linear transformation.

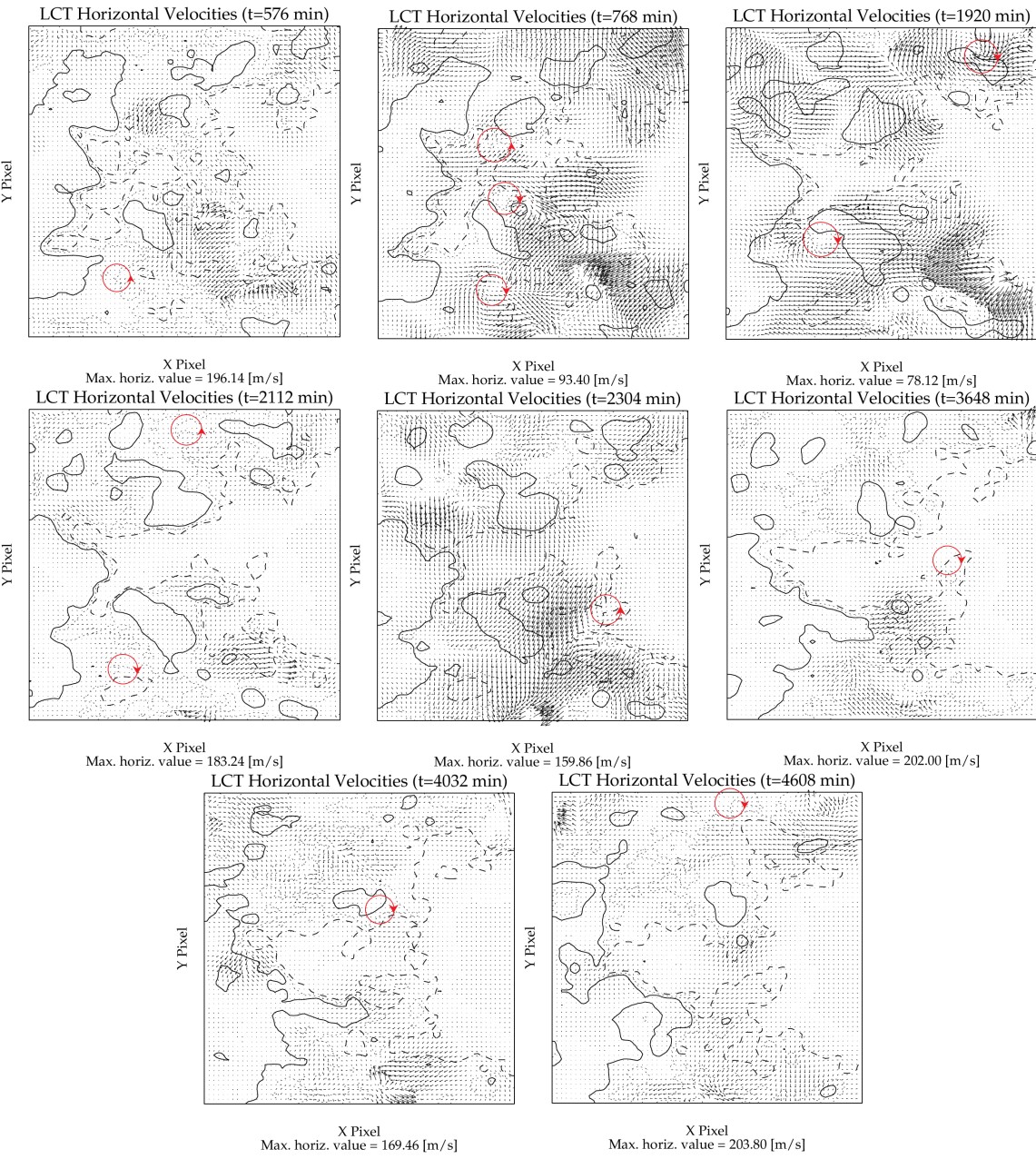

**Figure 4.** Location of the detected vortex flows in the 2D LCT flow fields obtained from the evolution of the magnetic structures around the PIL of a fully developed active region. The red circles indicate their location and the arrow indicate the direction of rotation.

Then, we sort a particular type of critical point called Attracting Focus which is associated to converging vortex flows. This procedure facilitates the visual identification of vortexes in the 2D photospheric flow fields and in our results we have identified a total of twelve converging vortexes in a period of 3.6 days. These converging vortexes show a clockwise preferred sense of

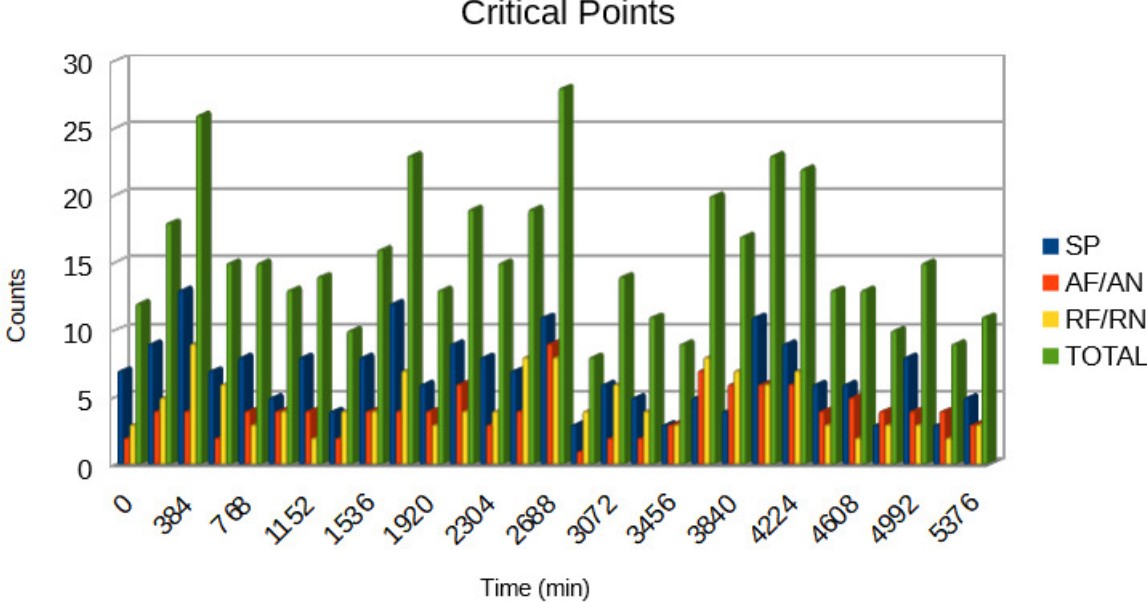

**Figure 5.** Noncumulative counts of critical points detected on the 2D LCT flows obtained from the evolution of the magnetic field around the PIL for the period of 3.6 days. The colors show their classification: blue-Saddle Point (SP), red-Attracting Node/Focus (AN/AF), yellow-Repelling Node/Focus (RN/RF).

rotation with approximately 67% of the cases. Attracting Focus is not the only kind of critical point detected, with the most common type being Saddle Points, with 213 detected in a 3.6 days period. These results reveal the complexity of the flow field around the PIL and suggests, together with previous results, that vortex flows are indeed a relatively common feature in the solar photosphere. By calculating the fractal dimension of the regions where the velocity is larger than a threshold value of $v = 23.6$ m/s, we obtain an average value of $D = 1.62 \pm 0.05$, which is very close to the values obtained for a homogeneous turbulence ($D = 5/3$). This reinforces the complexity of the flow around the PIL, suggesting that it presents a turbulent nature.

*Acknowledgements.* The authors would like to thank the anonimous referees for comments and suggestions, which helped to improve the quality of the article. This work was supported by the CNPq under the project 307653/2017-0. J. C. Santos would like to thank the CNPq for the PCI-E2 postdoc fellowship under the individual project 300890/2017-6.

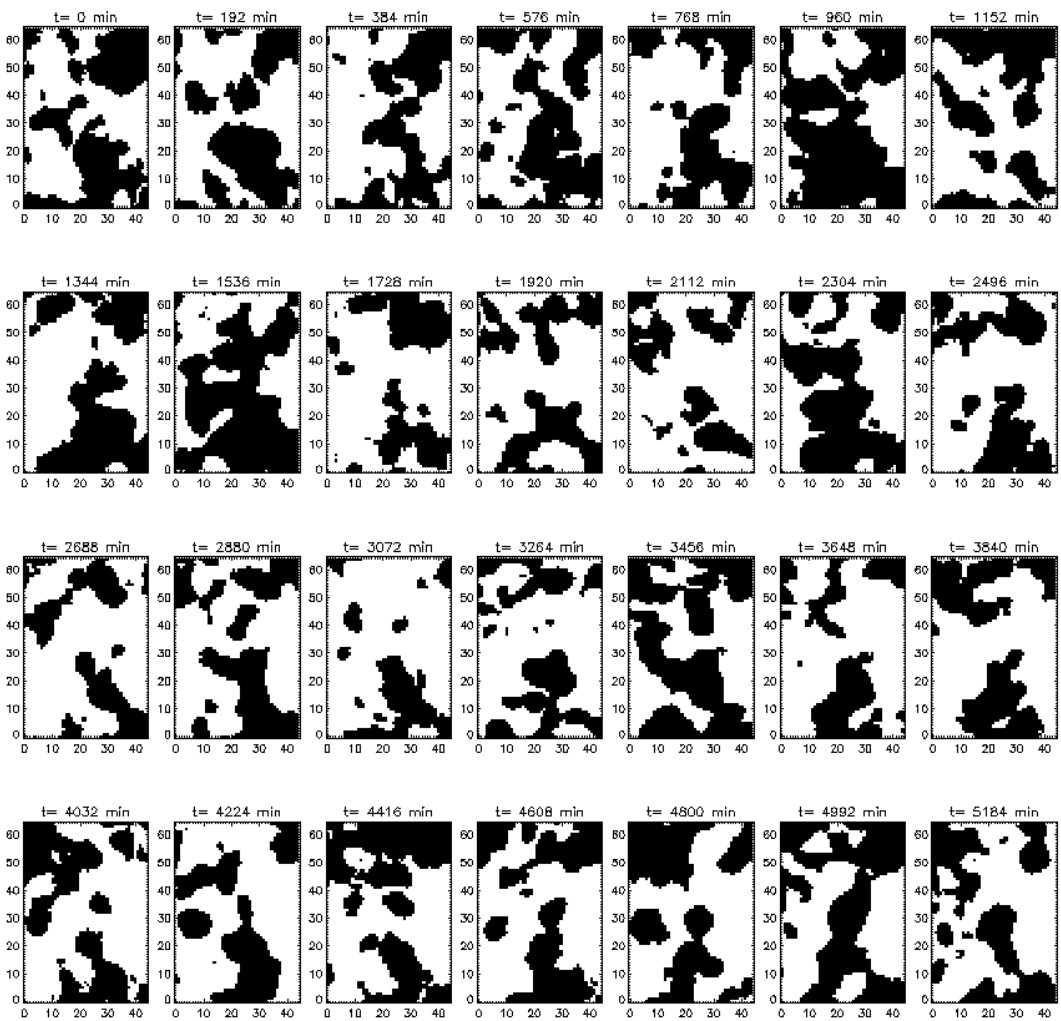

**Figure 6.** Time evolution of the regions where the velocity is above the threshold value of $v = 23.6$ m/s, shown in black, for a period of 3.6 days starting on December $31^{st}$ 2000 at 01:34:43 UT. The x and y axis show the spatial coordinates in pixel values, where each pixel corresponds approximately to 1.2 arcsec.

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

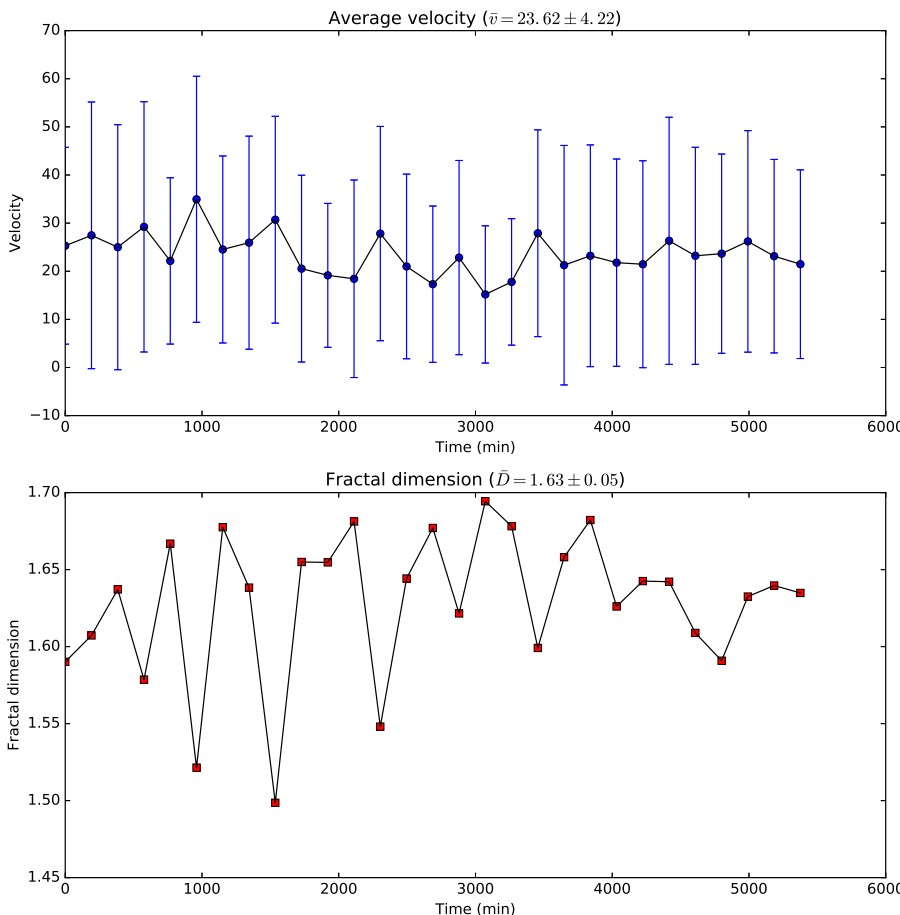

**Figure 7.** Temporal evolution of the average velocity (top panel) and fractal dimension (bottom panel) for the flow around the PIL.

Bi, Y., Jiang, Y., Yang, J., Hong, J., Li, H., Yang, B., and Xu, Z.: Observation of a reversal of rotation in a sunspot during a solar flare, Nature Communications, 7, 13798, https://doi.org/10.1038/ncomms13798, 2016.

Bonet, J. A., Márquez, I., Sánchez Almeida, J., Cabello, I., and Domingo, V.: Convectively Driven Vortex Flows in the Sun, The Astrophysical Journal, 687, L131, https://doi.org/10.1086/593329, 2008.

5    Bonet, J. A., Márquez, I., Sánchez Almeida, J., Palacios, J., Martínez Pillet, V., Solanki, S. K., del Toro Iniesta, J. C., Domingo, V., Berkefeld, T., Schmidt, W., Gandorfer, A., Barthol, P., and Knölker, M.: SUNRISE/IMaX Observations of Convectively Driven Vortex Flows in the Sun, The Astrophysical Journal, 723, L139–L143, https://doi.org/10.1088/2041-8205/723/2/L139, 2010.

Brandt, P. N., Scharmer, G. B., Ferguson, S., Shine, R. A., and Tarbell, T. D.: Vortex flow in the solar photosphere, , 335, 238–240, https://doi.org/10.1038/335238a0, 1988.

Brown, D. S., Nightingale, R. W., Alexander, D., Schrijver, C. J., Metcalf, T. R., Shine, R. A., Title, A. M., and Wolfson, C. J.: Observations of Rotating Sunspots from TRACE, , 216, 79–108, https://doi.org/10.1023/A:1026138413791, 2003.

Démoulin, P. and Berger, M. A.: Magnetic Energy and Helicity Fluxes at the Photospheric Level, Solar Physics, 215, 203–215, https://doi.org/10.1023/A:1025679813955, 2003.

Giagkiozis, I., Fedun, V., Scullion, E., Jess, D. B., and Verth, G.: Vortex Flows in the Solar Atmosphere: Automated Identification and Statistical Analysis, The Astrophysical Journal, 869, 169, https://doi.org/10.3847/1538-4357/aaf797, 2018.

Gopasyuk, O. S.: Rotation of sunspots in active region NOAA 10930, Advances in Space Research, 55, 937–941, https://doi.org/10.1016/j.asr.2014.09.005, 2015.

Hardersen, P. S., Balasubramaniam, K. S., and Shkolyar, S.: Chromospheric Mass Motions and Intrinsic Sunspot Rotations for NOAA Active
Regions 10484, 10486, and 10488 Using ISOON Data, The Astrophysical Journal, 773, 60, https://doi.org/10.1088/0004-637X/773/1/60, 2013.

Helman, J. and Hesselink, L.: Representation and display of vector field topology in fluid flow data sets, Computer, 22, 27–36, https://doi.org/10.1109/2.35197, 1989.

Hiremath, K. M. and Suryanarayana, G. S.: The flares associated with the abnormal rotation rates of the bipolar sunspots: Reconnection
probably below the surface, Astronomy & Astrophysics, 411, L497–L500, https://doi.org/10.1051/0004-6361:20031618, 2003.

Hiremath, K. M., Suryanarayana, G. S., and Lovely, M. R.: Flares associated with abnormal rotation rates: Longitudinal minimum separation of leading and following sunspots, Astronomy & Astrophysics, 437, 297–302, https://doi.org/10.1051/0004-6361:20042495, 2005.

Jiang, Y., Zheng, R., Yang, J., Hong, J., Yi, B., and Yang, D.: Rapid Sunspot Rotation Associated with the X2.2 Flare on 2011 February 15, The Astrophysical Journal, 744, 50, https://doi.org/10.1088/0004-637X/744/1/50, 2012.

Kato, Y. and Wedemeyer, S.: Vortex flows in the solar chromosphere. I. Automatic detection method, Astronomy & Astrophysics, 601, A135, https://doi.org/10.1051/0004-6361/201630082, 2017.

Kazachenko, M. D., Canfield, R. C., Longcope, D. W., and Qiu, J.: Sunspot Rotation, Flare Energetics, and Flux Rope Helicity: The Halloween Flare on 2003 October 28, The Astrophysical Journal, 722, 1539–1546, https://doi.org/10.1088/0004-637X/722/2/1539, 2010.

Kosovichev, A. G. and Zharkova, V. V.: Magnetic Energy Release and Transients in the Solar Flare of 2000 July 14, The Astrophysical
Journal, 550, L105–L108, https://doi.org/10.1086/319484, 2001.

Kusano, K., Bamba, Y., Yamamoto, T. T., Iida, Y., Toriumi, S., and Asai, A.: Magnetic Field Structures Triggering Solar Flares and Coronal Mass Ejections, The Astrophysical Journal, 760, 31, https://doi.org/10.1088/0004-637X/760/1/31, 2012.

Li, A. and Liu, Y.: Sunspot Rotation and the M-Class Flare in Solar Active Region NOAA 11158, Solar Physics, 290, 2199–2209, https://doi.org/10.1007/s11207-015-0745-5, 2015.

Liu, C., Xu, Y., Cao, W., Deng, N., Lee, J., Hudson, H. S., Gary, D. E., Wang, J., Jing, J., and Wang, H.: Flare differentially rotates sunspot on Sun's surface, Nature Communications, 7, 13104, https://doi.org/10.1038/ncomms13104, 2016.

Longcope, D. W.: Inferring a Photospheric Velocity Field from a Sequence of Vector Magnetograms: The Minimum Energy Fit, , 612, 1181–1192, https://doi.org/10.1086/422579, 2004.

Mandelbrot, B. B.: On the geometry of homogeneous turbulence, with stress on the fractal dimension of the iso-surfaces of scalars, Journal
of Fluid Mechanics, 72, 401–416, https://doi.org/10.1017/S0022112075003047, 1975.

Mandelbrot, B. B.: The Fractal Geometry of Nature, 1982.

Min, S. and Chae, J.: The Rotating Sunspot in AR 10930, Solar Physics, 258, 203–217, https://doi.org/10.1007/s11207-009-9425-7, 2009.

Moll, R., Cameron, R. H., and Schüssler, M.: Vortices, shocks, and heating in the solar photosphere: effect of a magnetic field, Astronomy & Astrophysics, 541, A68, https://doi.org/10.1051/0004-6361/201218866, 2012.

November, L. J. and Simon, G. W.: Precise proper-motion measurement of solar granulation, The Astrophysical Journal, 333, 427–442, https://doi.org/10.1086/166758, 1988.

Rempel, E. L., Chian, A. C. L., Beron-Vera, F. J., Szanyi, S., and Haller, G.: Objective vortex detection in an astrophysical dynamo, Monthly Notices of the Royal Astronomical Society: Letters, 466, L108–L112, https://doi.org/10.1093/mnrasl/slw248, 2017.

Ruan, G., Chen, Y., Wang, S., Zhang, H., Li, G., Jing, J., Su, J., Li, X., Xu, H., Du, G., and Wang, H.: A Solar Eruption Driven by Rapid Sunspot Rotation, The Astrophysical Journal, 784, 165, https://doi.org/10.1088/0004-637X/784/2/165, 2014.

Scherrer, P. H., Bogart, R. S., Bush, R. I., Hoeksema, J. T., Kosovichev, A. G., Schou, J., Rosenberg, W., Springer, L., Tarbell, T. D., Title, A.,
Wolfson, C. J., Zayer, I., and MDI Engineering Team: The Solar Oscillations Investigation - Michelson Doppler Imager, , 162, 129–188, https://doi.org/10.1007/BF00733429, 1995.

Schuck, P. W.: Local Correlation Tracking and the Magnetic Induction Equation, The Astrophysical Journal, 632, L53–L56, https://doi.org/10.1086/497633, 2005.

Schuck, P. W.: Tracking Vector Magnetograms with the Magnetic Induction Equation, The Astrophysical Journal, 683, 1134–1152,
https://doi.org/10.1086/589434, 2008.

Severnyi, A. B.: Nonstationary Processes in Solar Flares as a Manifestation of the Pinch Effect., , 2, 310, 1958.

Sharykin, I. N., Sadykov, V. M., Kosovichev, A. G., Vargas-Dominguez, S., and Zimovets, I. V.: Flare Energy Release in the Lower Solar Atmosphere near the Magnetic Field Polarity Inversion Line, The Astrophysical Journal, 840, 84, https://doi.org/10.3847/1538-4357/aa6dfd, 2017.

Simon, G. W. and Weiss, N. O.: Kinematic Modeling of Vortices in the Solar Photosphere, The Astrophysical Journal, 489, 960–967, https://doi.org/10.1086/304800, 1997.

Sudol, J. J. and Harvey, J. W.: Longitudinal Magnetic Field Changes Accompanying Solar Flares, The Astrophysical Journal, 635, 647–658, https://doi.org/10.1086/497361, 2005.

Suryanarayana, G. S., Hiremath, K. M., Bagare, S. P., and Hegde, M.: Abnormal rotation rates of sunspots and durations of associated flares,
Astronomy & Astrophysics, 580, A25, https://doi.org/10.1051/0004-6361/201423389, 2015.

Toriumi, S., Iida, Y., Bamba, Y., Kusano, K., Imada, S., and Inoue, S.: The Magnetic Systems Triggering the M6.6 Class Solar Flare in NOAA Active Region 11158, The Astrophysical Journal, 773, 128, https://doi.org/10.1088/0004-637X/773/2/128, 2013.

Vargas Domínguez, S., Palacios, J., Balmaceda, L., Cabello, I., and Domingo, V.: Spatial distribution and statistical properties of small-scale convective vortex-like motions in a quiet-Sun region, , 416, 148–154, https://doi.org/10.1111/j.1365-2966.2011.19048.x, 2011.

Vemareddy, P., Ambastha, A., and Maurya, R. A.: On the Role of Rotating Sunspots in the Activity of Solar Active Region NOAA 11158, The Astrophysical Journal, 761, 60, https://doi.org/10.1088/0004-637X/761/1/60, 2012.

Vemareddy, P., Cheng, X., and Ravindra, B.: Sunspot Rotation as a Driver of Major Solar Eruptions in the NOAA Active Region 12158, The Astrophysical Journal, 829, 24, https://doi.org/10.3847/0004-637X/829/1/24, 2016.

Wang, H., Ewell, Jr., M. W., Zirin, H., and Ai, G.: Vector magnetic field changes associated with X-class flares, The Astrophysical Journal,
424, 436–443, https://doi.org/10.1086/173901, 1994.

Wang, R., Liu, Y. D., Wiegelmann, T., Cheng, X., Hu, H., and Yang, Z.: Relationship Between Sunspot Rotation and a Major Solar Eruption on 12 July 2012, Solar Physics, 291, 1159–1171, https://doi.org/10.1007/s11207-016-0881-6, 2016.

Wang, S., Liu, C., Deng, N., and Wang, H.: Sudden Photospheric Motion and Sunspot Rotation Associated with the X2.2 Flare on 2011 February 15, The Astrophysical Journal, 782, L31, https://doi.org/10.1088/2041-8205/782/2/L31, 2014.

Wedemeyer-Böhm, S. and Rouppe van der Voort, L.: Small-scale swirl events in the quiet Sun chromosphere, Astronomy & Astrophysics, 507, L9–L12, https://doi.org/10.1051/0004-6361/200913380, 2009.

Welsch, B. T., Fisher, G. H., Abbett, W. P., and Regnier, S.: ILCT: Recovering Photospheric Velocities from Magnetograms by Combining the Induction Equation with Local Correlation Tracking, The Astrophysical Journal, 610, 1148–1156, https://doi.org/10.1086/421767, 2004.

Yan, X. L. and Qu, Z. Q.: Rapid rotation of a sunspot associated with flares, Astronomy & Astrophysics, 468, 1083–1088, https://doi.org/10.1051/0004-6361:20077064, 2007.

Yan, X.-L., Qu, Z.-Q., and Kong, D.-F.: Relationship between rotating sunspots and flare productivity, , 391, 1887–1892, https://doi.org/10.1111/j.1365-2966.2008.14002.x, 2008.

Yan, X.-L., Qu, Z.-Q., Xu, C.-L., Xue, Z.-K., and Kong, D.-F.: The causality between the rapid rotation of a sunspot and an X3.4 flare, Research in Astronomy and Astrophysics, 9, 596–602, https://doi.org/10.1088/1674-4527/9/5/010, 2009.

Yan, X. L., Qu, Z. Q., Kong, D. F., and Xu, C. L.: Sunspot Rotation, Sigmoidal Filament, Flare, and Coronal Mass Ejection: The Event on

2000 February 10, The Astrophysical Journal, 754, 16, https://doi.org/10.1088/0004-637X/754/1/16, 2012.

Zhu, C., Alexander, D., and Tian, L.: Velocity Characteristics of Rotating Sunspots, Solar Physics, 278, 121–136, https://doi.org/10.1007/s11207-011-9923-2, 2012.