# Peer review of "Converging photospheric vortex flows close to the polarity inversion line of a fully emerged active region"

_Annales Geophysicae, 2019_

## Referee Comment (RC1) · Anonymous Referee #1 · 8 Apr 2019

Referee Comments:

Using the tracked flow velocity field from the time-sequence magnetograms, the authors found vortex flow patterns surrounding the polarity inversion line and claims the turbulent nature of the plasma by a geometric method. I believe this is an attempt to show the turbulence nature of the flow on the Sun. I have the following comments which may help improve the manuscript.

Abstract: "Using this method. . ." this sentence is not in continuation with the earlier, confusing which method

Section 2: cadence & resolution of the MDI B_LOS needs to be mentioned. It is related

to the discussion of the LCT method employed on them.

P.4, paragraph "To determine the velocity field...", FLCT is a method is valid for the intensity based images. However, magnetic fields on the sun evolve according to the magnetic induction equation, so LCT is to be modified accounting the induction equation. You may refer to Schuck (2005) for the differential affine velocity estimator (DAVE) technique. However, I would suggest to check the detection of the same critical points in the flow patters derived from DAVE. I am sure that the flow patterns of vortical nature would exhibit enhanced curvature with DAVE. This needs to be properly discussed. Also please mention the size of the apodising window used.

I have some issue with the cadence of the B_los used. A cadence of 192minutes is too high to track flow velocity and you would loss the vortical patterns. Assuming a 0.5 km/s velocity, in 192 minutes, the motion is around 8 arcsec. Then the critical points found with the velocity field in hand are of major concern. How about using HMI magnetograms at a higher cadence ?

P.7, I would suggest to write a brief description on the box-counting method. How does it relates to the kolmogorov power law. This information is needed from a new reader perspective.

Regarding the critical points detection, I have a concern on the threshold of the magnetic field. Usually, the flow velocity is somewhat noisy in the weak-field regions, ofcourse that is the key issue for the discussion of the turbulent nature of the plasma. Then the identified critical points, Figure 6, in the weak field regions especially in the PIL regions are subjective.Please provide a solid justification.

p.7, "Since the fully developed turbulence consists of a hierarchy...", it could be, but in the presence of the magnetic field, it can be quenched, then there is point to think why the vortices are found only at certain points. I mean to ask, what about the power law at the places other than critical points. Generally, turbulence is present every on the sun, then what kind of power is expected for example in some what magnetic field
regions. Is the geometrical method used still applicable there?

---

## Referee Comment (RC2) · Anonymous Referee #2 · 20 Apr 2019

General comments

The article entitled "Photospheric vortex flows close to the polarity inversion line of a fully emerged active region" presents a new approach to evidence the occurrence of vortices in the magnetic flux fields in solar active regions. In the methodology presented, the occurrence of vortices is associated with the presence of critical points. And they are evidenced from the evolution of the component of the photospheric magnetic field in the line of sight in a region near the polarity inversion line (PIL) of a solar active region fully established, that is, that it was not in an emergency or evanescence stage. Both the original approach and the results (somehow expected) suggesting the

turbulent nature of the flow around the polarity inversion line are interesting and, in my point of view, deserve to be shared through the publication of the article.

The manuscript, in general, is written well, the data and the methodology are clearly presented. Just like the results. Anyway, I highlight below some points and questions about the text, with the intention of trying to contribute to improve the article. Some comments are only suggestions, which authors should accept if they think they are appropriate.

Section 2 Data and Methodology

In this section, the choice of the active region NOAA 9289, located at the southern solar hemisphere, and the companions between December 31, 2000 and January 3, 2001, is cited. As the magnetogram (MDI) taken on January 2, 2001 and shown in figure 1, this region is located very close to the center of the Sun.

Questions: Does the methodology used necessarily require that the analyzed region to be close to the center of the solar disk, to avoid projection effects?

Does the positional variation of the active region during the 3.6 days whose evolution accompanied carry some implication of the determinations of the evolution of the magnetic field along the line of sight (BLOS)? In addition, it is mentioned that a range of 192 min. was considered for the BLOS data, in this time interval, the variations in flows and speeds may be significant for the establishment of the obtained vorticity patterns? What considerations, implications, approximations (if any) should be made in the case of an active region close to the solar slime, for example? The authors should comment on all this aspects.

Even if these aspects or requirements are not problems for application of the method, the authors could perhaps comment on this. If there is a limitation of time interval (days), considering the displacement of the active region to follow the evolution of the active region for evidence of vortices.

In the beginning (first sentence) of page 4, the authors describe that at time t = 1920 min. begins to form a negative polarity region (N1), connected to the active region main negative polarity, and two small positive polarity regions, one northern of the negative polarity (P1) and other southern (P2). However, in the previous frame shown in figure 2 for t = 1728 min. (or even for t = 1532 min.) these same N1, P1 and P2 regions are already identifiable. My question: what criterion (visual only?) was used to identify these instants and stages (coalescence of polarity fragmentation and establishment / structuring of regions with well defined polarities) from the magnetogram images?

Page 7 Figure 3: Colors and symbols (asterisks) used, especially yellow, to show the location of the critical points (asterisks) (blue: Saddle Point, red: Attracting Node / Focus, yellow: Repelling Node / Focus) are small and difficult to see without magnification.

Section 2.1

I suggest using the same notation to denote the components x and y (x, y). In the equations and matrices they are typed in italics and in the text they are not.

In the last sentence of page 5, the authors state that "Critical points are the salient features of a flow pattern". This statement seems somewhat vague, must be better clarified (based on what consideration or criterion) or referenced.

Section results

Suggestion: Presented sequentially the figures 4 and 5 (top and bottom panels), according to the results they want to present. In the first paragraph of the section – results, the authors cite that they first investigated the fractal dimension of flowing 2D structures, the results of which appear only in the bottom panel of Figure 5. However, the authors cite the Figure 5 (velocity) before the Figure 4.

In my opinion, it might be more coherent to present the results sequentially. First the velocity values (figure 5 top), then the evolution of the regions which present velocity

above that calculated from the velocities of figure 5 (figure 4). Finally, the fractal dimensions (Figure 5 bottom). For readers, it may not be clear which parameters were determined from which others. Maybe it separates them in 3 figures (4, 5 and 6).

They were select only the critical points classified as Attracting Focus, which represent vortices that converge to this particular point. Were identified any points scored as Repelling Focus? They represent vortices as well, but diverging. Can do these critical points also contribute to the nature of plasma turbulence?

And on the saddle points, in the conclusion the authors mention that they are the types of critical points more common. However, in the present work, they do not mention how many saddle points were detected and what they represent in this analysis of the dynamics of the plasma and the flux of the photospheric magnetic field. However, in the present work they do not mention how many saddle points was detected and what they represent in this analysis of the dynamics of the plasma and the flux of the photospheric magnetic field.

On the determination of velocities. Figure 5 (bottom) shows the velocity values. It may be noted that the error bars are large (on the order of $\sim 100\%$ (!!).) Discuss the implications of these errors in identifying critical points and vorticity.

The values of the fractal dimension also have variations shown in Figure 5 (bottom). How to interpret these fluctuations in the fractal dimension during the evolution of the region around the PIL?

Figure 7 (page 11). Abbreviations of the critical points presented in the caption of figure 7 are not in the text. I suggest including it in the caption.

Conclusions

The complex and turbulent nature of the configuration and evolution of the photosphere magnetic field associated with the active regions is relatively well established. The results presented in this paper, even if only for one case analyzed, reinforce this previous

evidence in the case of the flow around the PIL. It is give emphasis to the innovative methodology presented.

---

## Referee Comment (RC3) · Anonymous Referee #3 · 8 May 2019

**1 General comments**

This article deals with the topic of vorticity computed on velocity field in the solar photosphere. The chosen topic is quite relevant, since this magnitude can have different applications in solar physics (as the stated for flares) among others (see in Specific comments). The idea of linking vorticity to fractal dimension is interesting. I suggest to perform some other comparisons on another magnetic regions, in order to show whether this fractal dimension is really dependent on the region taken (PIL) or it would be similar/different to other solar magnetic regions. It would be important to clarify how the fractal dimension would be/would not be dependent of the local correlation tracking

(LCT) parameters (spatial and temporal).

This paper is clearly written, with good potential, which can be improved with some of the extra analyses suggested below. However, I have a number of questions and concerns, that hopefully can be clarified.

**2 Specific comments**

a. Page 2, line 7: In Bonet et al., 2008, it is said that “how that the vortexes are indeed associated to the occurrence of bright points”. It is actually the other way round, the bright points are indeed used as vortex tracers.

b. Page 2, line 7: However, strong flares (M- or X-class) are usually associated to rapid (abnormal) sunspot rotation. I think it is more precise “However, strong flares (M- or X-class) are **sometimes** associated to rapid (abnormal) sunspot rotation”, since the focus is on rotation but the main mechanism may be shear motions. Vorticity is very important, not only in the context for flares, but in different solar scenarios (e.g., the authors can check the relationship of vorticity and internal waves in Vigeesh et al., 2017).

c. Fig 2. In this figure, the polarity N1 looks more like deforming, while the polarity P1 looks like protruding into N1. For some description on polarity protrusion and their role in flares, please refer to Kusano et al. (2012); Toriumi et al. (2013).

d. In page 5, line 1, please mention why FLCT is preferred over LCT, and over other more appropriate methods for magnetograms like LCT with induction equation (i.e., DAVE-4VM, Schuck, 2008).

e. The chosen cadence (192 min) probably will lead to very small values on the surface velocity field (which it seems the case), which makes the vortices detection

more complicated (meaning “vortices” on typical granulation times, as minutes, and then allowing the long-lasting vortices to be detected). The cadence for LCT usually is adjusted to the structures one desires to track. How is this considered in this work?

f. *Importantly, the method described in Section 2.1 is similar to that developed in Kato & Wedemeyer (2017) (see references therein for their basis, as Chong et al. 1990). Please cite also this work, and it can be used for comparison. Also another very recent method is explained in Giagkiozis et al. (2018) and some references therein.*

g. Page 6, line 20: some percentages on false positives and missing events would improve the quality of the work.

h. Since LCT is very dependent on cadence and spatial sampling, one wonders how the result on fractal dimension would be with HMI data (0.6” pixel$^{-1}$), different cadence (shorter than 192 min), and around PIL/around a one-polarity region. This work would really improve by adding these extra analyses and re-computing the $D$ dimension.

i. Page 9, line 2. Are these vortices percentage dependent on the solar hemisphere? May they be dependent on the PIL? The extra analyses (as inside/outside the PIL) can contribute also to this particular point and moreover, to the whole work relevance.

**3 Technical comments**

1. In the abstract (page 1, line 4), it is said “eigenvalues of the Jacobian matrix of the linear transformation”. Of which magnitude? One guesses that it is the surface velocity field.

2. Page 1, line 22: “In the quiet Sun convective flows concentrate magnetic fields in the downdraft region”. Please change to plural, regions.
3. Please mention the cadence and spatial sampling of MDI in the data description paragraph (starting in page 2, line 29). The spatial sampling and cadence only appears when explaining Fig.1. Please mention also that they are full disk MDI data, since potential readers may be not fully familiar with solar imaging datasets.
4. Please add units in Figure 1. Are these arcsecs?
5. Equations: A hyphen over the letter is a bit misleading, since it reminds to a vector. Probably other symbol would be a better choice.
6. Page 6, line 18: “The solid (dashed) contour line indicate the regions where BLOS assumes the value of +100 G (-100 G). Probably is better explained as “The solid (dashed) contour line indicate the regions where BLOS equals the value of +100 G (-100 G)”
7. Page 6, line 22: “The the identification of the critical points (LIC)” probably can be rephrased as: hereafter, LICs. What “LIC” does stand for in this work? Is it ‘line integral convolution’, as in Kato & Wedemeyer (2017)?
8. Please add maximum and minimum values for the units in Figure 3.
9. Page 7, line 9, section Results: please detail how the fractal dimension is computed in this case.
10. Page 7, line 16: Please explain how is resampled (what was the original size of the image which is resampled to 128x128?
11. Please add units in Figure 4.
12. Please detail in the text the content of Figure 7. Are these counts non-cumulative? Are they computed every time step?

*References*

Giagkiozis, I., Fedun, V., Scullion, E., Jess, D. B., & Verth, G. 2018, ApJ., 869, 169 DOI: https://doi.org/10.3847/1538-4357/aaf797

Kato, Y., & Wedemeyer, S. 2017, A&A, 601, A135, DOI: https://doi.org/10.1051/0004-6361/201630082

Kusano, K., Bamba, Y., Yamamoto, T. T., et al. 2012, ApJ., 760, 31, DOI: https://doi.org/10.1088/0004-637X/760/1/31

Vigeesh, G., Steiner, O., Calvo, F., & Roth, M. 2017, Mem. S.A. It., 88, 54

Schuck, P. W. 2008, ApJ., 683, 1134

Toriumi, S., Iida, Y., Bamba, Y., et al. 2013, ApJ., 773, 128 , DOI:https://doi.org/10.1088/0004-637X/773/2/128

---

## Author Comment (AC1) · 4 Jun 2019

Dear Referee,

thank you very much for your comments and questions, they certainly helped to improve the quality of the article. Below I have tried to answer most of your comments and questions.

Abstract: "Using this method. . ." this sentence is not in continuation with the earlier, confusing which method

Answer: The sentence was rewritten in the revised version of the manuscript.

—————————————————

Section 2: cadence & resolution of the MDI B_LOS needs to be mentioned. It is related to the discussion of the LCT method employed on them.

Answer: The information about cadence & resolution of the MDI B_LOS was inserted in the revised version of the manuscript.

—————————————————

P.4, paragraph "To determine the velocity field. . .", FLCT is a method is valid for the intensity based images. However, magnetic fields on the sun evolve according to the magnetic induction equation, so LCT is to be modified accounting the induction equation. You may refer to Schuck (2005) for the differential affine velocity estimator (DAVE) technique. However, I would suggest to check the detection of the same critical points in the flow patters derived from DAVE. I am sure that the flow patterns of vortical nature would exhibit enhanced curvature with DAVE. This needs to be properly discussed. Also please mention the size of the apodising window used.

Answer: LCT/FLCT is the easiest method to implement and requires only the information about the temporal evolution of the LOS component of the magnetic field. This was the main reasons the method was used to obtain the horizontal velocity. In the revised version of the manuscript we discuss the weakness of the LCT/FLCT method and the corrections suggested by different methods available (ILCT/MEF/DAVE). The information about the size of the apodizing window is already mentioned in the revised version of the manuscript.

—————————————————

I have some issue with the cadence of the B_los used. A cadence of 192minutes is too high to track flow velocity and you would loss the vortical patterns. Assuming a 0.5 km/s velocity, in 192 minutes, the motion is around 8 arcsec. Then the critical points found with the velocity field in hand are of major concern. How about using HMI

magnetograms at a higher cadence ?

Answer: The reason we have used MDI data is that we wanted to investigate a fully emerged active region that presented rotation in at least one of the main polarities to cause shear in the PIL. The selected active region has been already described in Zhu 2012 and presented those characteristics. We have used 96 min full disk MDI data and have applied LCT/FLCT to cadences 96, 192, 288, . . . The selected cadence presented the best result in terms of velocity field obtained using LCT/FLCT and vortex detection.

—————————————————

P.7, I would suggest to write a brief description on the box-counting method. How does it relates to the kolmogorov power law. This information is needed from a new reader perspective.

Answer: A description of the box-counting method and its relation to turbulence, with references, is presented in the revised version of the manuscript.

—————————————————

Regarding the critical points detection, I have a concern on the threshold of the magnetic field. Usually, the flow velocity is somewhat noisy in the weak-field regions, of course that is the key issue for the discussion of the turbulent nature of the plasma. Then the identified critical points, Figure 6, in the weak field regions especially in the PIL regions are subjective.Please provide a solid justification.

Answer: This is a difficulty associated to the analysis of the flow field around the polarity inversion line. The noise level of MDI full disk 96 min is about 7.6 gauss and the method for detection is sensitive to the velocity far from the location of the critical point. This give us some confidence that the critical points that are large enough and are captured at least in part in the motion of stronger magnetic field structures are actually there. However, there is no way to be 100% sure about the small vortexes detected in noise level magnetic field regions.

————————————————

p.7, "Since the fully developed turbulence consists of a hierarchy. . .", it could be, but in the presence of the magnetic field, it can be quenched, then there is point to think why the vortices are found only at certain points. I mean to ask, what about the power law at the places other than critical points. Generally, turbulence is present every on the sun, then what kind of power is expected for example in some what magnetic field regions. Is the geometrical method used still applicable there?

Answer: Abramenko V. I. has investigated the multifractal nature of the magnetic field of active regions and studied the power spectrum of the magnetic field, showing that it presents turbulent characteristics (a linear region in the power spectrum with Kolmogorov type power law). The geometrical method is already used to study the multifractal nature of active regions and we plan to use it to further investigate the evolution of active regions under turbulent diffusion, after emergence.

Please also note the supplement to this comment:
https://www.ann-geophys-discuss.net/angeo-2019-33/angeo-2019-33-AC1-supplement.pdf

---

## Author Comment (AC2) · 4 Jun 2019

Dear referee,

thank you very much for your comments and questions, they certainly helped to improve the quality of the article. Below I have tried to answer most of your comments and questions.

Section 2 Data and Methodology In this section, the choice of the active region NOAA 9289, located at the southern solar hemisphere, and the companions between December 31, 2000 and January 3, 2001, is cited. As the magnetogram (MDI) taken on

January 2, 2001 and shown in figure 1, this region is located very close to the center of the Sun. Questions: Does the methodology used necessarily require that the analyzed region to be close to the center of the solar disk, to avoid projection effects?

Answer: For active regions far from the disk center (closer to limb) the validity of the assumption that B_LOS is equal to the perpendicular component of the magnetic field is not true. The derivation of the velocity usually assumes that we are following the temporal evolution of the normal component of the magnetic field. So, in this case the B_LOS should be corrected for projection effects in order that we may assume the equality of the normal component and B_LOS.

———————————————-

Does the positional variation of the active region during the 3.6 days whose evolution accompanied carry some implication of the determinations of the evolution of the magnetic field along the line of sight (BLOS)?

Answer: The effects of the variation in position of the active region during the 3.6 days are considered small enough that they could be disregarded in this case.

———————————————-

In addition, it is mentioned that a range of 192 min. was considered for the BLOS data, in this time interval, the variations in flows and speeds may be significant for the establishment of the obtained vorticity patterns?

Answer: We have used the full disk 96 min cadence data from MDI. We calculated the velocity field for the intervals 96min, 192min, 288 min, . . . the best results were obtained for the 192 min. For the 96 min the LCT/FLCT method was not able to capture the vortex patterns with the apodizing window selected.

———————————————-

What considerations, implications, approximations (if any) should be made in the case

of an active region close to the solar slime, for example?

Answer: As far as I know, for regions very close to the solar limb the method has problems to be implemented and is not useful.

————————————————-

In the beginning (first sentence) of page 4, the authors describe that at time t = 1920 min. begins to form a negative polarity region (N1), connected to the active region main negative polarity, and two small positive polarity regions, one northern of the negative polarity (P1) and other southern (P2). However, in the previous frame shown in figure 2 for t = 1728 min. (or even for t = 1532 min.) these same N1, P1 and P2 regions are already identifiable. My question: what criterion (visual only?) was used to identify these instants and stages (coalescence of polarity fragmentation and establishment structuring of regions with well defined polarities) from the magnetogram images?

Answer: The description of the evolution of the magnetic field around the polarity inversion line was qualitative and based in visual criterion. The text of the corrected version of the manuscript was modified to make this point clear. Page 7 Figure 3: Colors and symbols (asterisks) used, especially yellow, to show the location of the critical points (asterisks) (blue: Saddle Point, red: Attracting Node / Focus, yellow: Repelling Node / Focus) are small and difficult to see without magnification.

————————————————-

Answer: The figure was modified in the corrected version of the manuscript in order to improve the quality.

I suggest using the same notation to denote the components x and y (x, y). In the equations and matrices they are typed in italics and in the text they are not.

Answer: The notation was modified in the corrected version of the manuscript.

————————————————-

In the last sentence of page 5, the authors state that "Critical points are the salient features of a flow pattern". This statement seems somewhat vague, must be better clarified (based on what consideration or criterion) or referenced.

Answer: What we wanted to say here is that critical points have more significance than just points where the velocity vanishes. We have modified the corrected version of the manuscript and interpreted the critical points as fixed points of a map in a dynamical system, where if we know their location and their type we can predict the orbit of a particle around the position of the critical points.

——————————————————-

Suggestion: Presented sequentially the figures 4 and 5 (top and bottom panels), according to the results they want to present. In the first paragraph of the section results, the authors cite that they first investigated the fractal dimension of flowing 2D structures, the results of which appear only in the bottom panel of Figure 5. However, the authors cite the Figure 5 (velocity) before the Figure 4.

In my opinion, it might be more coherent to present the results sequentially. First the velocity values (figure 5 top), then the evolution of the regions which present velocity above that calculated from the velocities of figure 5 (figure 4). Finally, the fractal dimensions (Figure 5 bottom). For readers, it may not be clear which parameters were determined from which others. Maybe it separates them in 3 figures (4, 5 and 6).

Answer: We have modified the order of presentation of the results in the revised version of the manuscript.

——————————————————-

They were select only the critical points classified as Attracting Focus, which represent vortices that converge to this particular point. Were identified any points scored as Repelling Focus? They represent vortices as well, but diverging. Can do these critical points also contribute to the nature of plasma turbulence?

Answer: There are both, attracting and repelling focus, present in the 2D flow field around the PIL. We focus on attracting focus since this kind of vortex flow can bring together opposite polarities or increase the amplitude of the unipolar field at the same time that it twists the field lines. Landau-Hopf theory of turbulence requires that the flow increases the modes present in the Fourier decomposition, but nothing is told about the direction of the vortexes.

———————————————

And on the saddle points, in the conclusion the authors mention that they are the types of critical points more common. However, in the present work, they do not mention how many saddle points were detected and what they represent in this analysis of the dynamics of the plasma and the flux of the photospheric magnetic field. However, in the present work they do not mention how many saddle points was detected and what they represent in this analysis of the dynamics of the plasma and the flux of the photospheric magnetic field.

Answer: We included the total number of detected saddle points in the revised version of the manuscript.

———————————————

Figure 7 (page 11). Abbreviations of the critical points presented in the caption of figure 7 are not in the text. I suggest including it in the caption.

Answer: We have included the description of the abbreviations in the caption of the figure in the revised version of the manuscript.

———————————————

The values of the fractal dimension also have variations shown in Figure 5 (bottom). How to interpret these fluctuations in the fractal dimension during the evolution of the region around the PIL?

Answer: Changes in the value of the fractal dimension imply changes in the self similarity of the analyzed structures. This could be related with a change in the physical process occurring in the flow. But it is difficult to make this affirmation for the flow we have analyzed.

Please also note the supplement to this comment:
https://www.ann-geophys-discuss.net/angeo-2019-33/angeo-2019-33-AC2-supplement.pdf
* * *
[Figure]

**Supplement:**

[revised manuscript text omitted]

---

## Author Comment (AC3) · 4 Jun 2019

Dear Referee,

thank you very much for your comments and questions, they certainly helped to improve the quality of the article. Below I have tried to answer most of your comments and questions.

Page 2, line 7: In Bonet et al., 2008, it is said that "how that the vortexes are indeed associated to the occurrence of bright points". It is actually the other way round, the bright points are indeed used as vortex tracers.

[Figure]

Answer: this sentence was modified in the revised version of the manuscript.
* * *
Page 2, line 7: However, strong flares (M- or X-class) are usually associated to rapid (abnormal) sunspot rotation. I think it is more precise "However, strong flares (M- or X-class) are sometimes associated to rapid (abnormal) sunspot rotation", since the focus is on rotation but the main mechanism may be shear motions. Vorticity is very important, not only in the context for flares, but in different solar scenarios (e.g., the authors can check the relationship of vorticity and internal waves in Vigeesh et al., 2017).

Answer: thank you very much for the reference. The sentence was modified in the revised version of the manuscript.
* * *
Fig 2. In this figure, the polarity N1 looks more like deforming, while the polarity P1 looks like protruding into N1. For some description on polarity protrusion and their role in flares, please refer to Kusano et al. (2012); Toriumi et al. (2013).

Answer: thank you very much for the references and for this comment. It is very difficult to notice all the details of the evolution visually. I have inserted a sentence describing what you have mentioned.
* * *
In page 5, line 1, please mention why FLCT is preferred over LCT, and over other more appropriate methods for magnetograms like LCT with induction equation (i.e., DAVE-4VM, Schuck, 2008).

Answer: the LCT/FLCT method was used due its simplicity to implement and since it requires only the LOS component of the magnetic field. We included a discussion of more appropriated methods in the revised version of the manuscript and will try to

implement the DAVE method for using in the next works.

————————————————————

The chosen cadence (192 min) probably will lead to very small values on the surface velocity field (which it seems the case), which makes the vortices detection more complicated (meaning "vortices" on typical granulation times, as minutes, and then allowing the long-lasting vortices to be detected). The cadence for LCT usually is adjusted to the structures one desires to track. How is this considered in this work?

Asnwer: we have used full disk 96min cadence MDI data to study this particular active region. We have applied the LCT/FLCT method to different cadences 96, 192, 288, . . . and the best result in terms of velocity flow and vortex detection was the presented cadence. The vortex detection method seems to work reasonably well independent of the velocity amplitude, but this should be checked. For future analysis we will use HMI data, which has better spatial and temporal resolution.

————————————————————

Importantly, the method described in Section 2.1 is similar to that developed in Kato & Wedemeyer (2017) (see references therein for their basis, as Chong et al. 1990). Please cite also this work, and it can be used for comparison. Also another very recent method is explained in Giagkiozis et al. (2018) and some references therein.

Answer: thank you very much for the references. I have included them into this work and also the one from rempel (2017).

————————————————————

Page 6, line 20: some percentages on false positives and missing events would improve the quality of the work.

Answer: I have included a percentage which represents the accuracy of the method (about 70%) in finding and classifying the critical points correctly.
* * *
Since LCT is very dependent on cadence and spatial sampling, one wonders how the result on fractal dimension would be with HMI data (0.6" pixel $-1$ ), different cadence (shorter than 192 min), and around PIL/around a one-polarity region. This work would really improve by adding these extra analyses and re-computing the D dimension.

Answer: thank you very much for your suggestion. We are starting the analysis of fractal dimension of magnetic structures in the solar atmosphere and, different of other works that analyze the fractal dimension of the whole active region, we would like to investigate in more detail the PIL of different active regions related to flares. However, we are still understanding the meaning of the fractal dimension in the case of the Sun. I would think that the increase in spatial resolution of the data would not change this value since it is related to the self similarity of the geometrical structure.
* * *
Page 9, line 2. Are these vortices percentage dependent on the solar hemisphere? May they be dependent on the PIL? The extra analyses (as inside/outside the PIL) can contribute also to this particular point and moreover, to the whole work relevance.

Answer: for vortexes detected outside active regions the percentage depends on the solar hemisphere. However, I did not read any work on these percentages for vortexes detected on the PIL or around an active region. That is a good topic for investigation in a future work.
* * *
In the abstract (page 1, line 4), it is said "eigenvalues of the Jacobian matrix of the linear transformation". Of which magnitude? One guesses that it is the surface velocity field.

Answer: the eigenvalues obtained were not far from unit when they were not zero.

———————————————

Page 1, line 22: "In the quiet Sun convective flows concentrate magnetic fields in the downdraft region". Please change to plural, regions.

Answer: this error was corrected in the revised version of the manuscript.

———————————————

Please mention the cadence and spatial sampling of MDI in the data description paragraph (starting in page 2, line 29). The spatial sampling and cadence only appears when explaining Fig.1. Please mention also that they are full disk MDI data, since potential readers may be not fully familiar with solar imaging datasets.

Answer: the information was included in the revised version of the manuscript.

———————————————

Please add units in Figure 1. Are these arcsecs?

Answer: the units are in pixels. I have included in the caption of the figure.

———————————————

Equations: A hyphen over the letter is a bit misleading, since it reminds to a vector. Probably other symbol would be a better choice.

Answer: the symbols were modified in the revised version of the manuscript.

———————————————

Page 6, line 18: "The solid (dashed) contour line indicate the regions where BLOS assumes the value of +100 G (-100 G). Probably is better explained as "The solid (dashed) contour line indicate the regions where BLOS equals the value of +100 G (-100 G)"

Answer: the sentence was modified in the revised version of the manuscript.
* * *
Page 6, line 22: "The the identification of the critical points (LIC)" probably can be rephrased as: hereafter, LICs. What "LIC" does stand for in this work? Is it 'line integral convolution', as in Kato & Wedemeyer (2017)?

Answer: yes, LIC stands for line integral convolution. The sentence was modified in the revised version of the manuscript.
* * *
Please add maximum and minimum values for the units in Figure 3.

Answer: The minimum values are always zero.
* * *
Page 7, line 9, section Results: please detail how the fractal dimension is computed in this case.

Answer: a section was included in the revised version of the manuscript describing the fractal dimension computation method.
* * *
Page 7, line 16: Please explain how is resampled (what was the original size of the image which is resampled to 128x128?

Answer: the original velocity field had 45x65 and was resampled using nearest-neighbor interpolation.
* * *
Please detail in the text the content of Figure 7. Are these counts noncumulative? Are they computed every time step?

Answer: the counts are noncumulative and are computed every time step. We included

this information in the revised version of the manuscript.

Please also note the supplement to this comment:
https://www.ann-geophys-discuss.net/angeo-2019-33/angeo-2019-33-AC3-
supplement.pdf

**Supplement:**

[revised manuscript text omitted]